# A Comparative Study of Dynamic Light and X-ray Scatterings on Micelles of Topological Polymer Amphiphiles

**DOI:** 10.3390/polym10121347

**Published:** 2018-12-05

**Authors:** Brian J. Ree, Jongchan Lee, Yusuke Satoh, Kyungho Kwon, Takuya Isono, Toshifumi Satoh, Moonhor Ree

**Affiliations:** 1Graduate School of Chemical Sciences and Engineering, Hokkaido University, Sapporo 060-8628, Japan; brianree@poly-bm.eng.hokudai.ac.jp (B.J.R.); ysatoh@poly-bm.eng.hokudai.ac.jp (Y.S.); 2Department of Chemistry, Division of Advanced Materials Science, Polymer Research Institute, and Pohang Accelerator Laboratory, Pohang University of Science and Technology, Pohang 37673, Korea; gleeprince@postech.ac.kr (J.L.); kkh85@postech.ac.kr (K.K.); 3Faculty of Engineering, Hokkaido University, Sapporo 060-8628, Japan; isono@eng.hokudai.ac.jp

**Keywords:** amphiphilic cyclic diblock copolymer, amphiphilic linear diblock copolymer, micelle size, size distribution, micellar structure, dynamic light scattering, synchrotron X-ray scattering

## Abstract

Micelles were prepared in organic solvents by using three topological polymer amphiphiles: (i) *cyclic* poly(*n*-decyl glycidyl ether-*block*-2-(2-(2-methoxyethoxy)ethoxy)ethyl glycidyl ether) (*c*-PDGE-*b*-PTEGGE) and (ii) its *linear* analogue (*l*-PDGE-*b*-PTEGGE); (iii) *linear* poly(6-phosphorylcholinehexylthiopropyl glycidyl ether-*block*-*n*-dodecanoyl glycidyl ether) (*l*-PPCGE-*b*-PDDGE). For the individual micelle solutions, the size and distribution were determined by dynamic light scattering (DLS) and synchrotron X-ray scattering analyses. The synchrotron X-ray scattering analysis further found that *c*-PDGE-*b*-PTEGGE forms oblate ellipsoidal micelle in an ethanol/water mixture, *l*-PDGE-*b*-PTEGGE makes prolate ellipsoidal micelle in an ethanol/water mixture, and *l*-PPCGE-*b*-PDDGE forms cylindrical micelle in chloroform. This comparative study found that there are large differences in the size and distribution results extracted by DLS and X-ray scattering analyses. All possible factors to cause such large differences are discussed. Moreover, a better use of the DLS instrument with keeping its merits is proposed.

## 1. Introduction

Nanomaterials are widely adopted as key components in the development of new advanced technologies. In general, nanomaterials have dimensions between small molecules and bulk materials and exhibit very different properties from their counterparts. Over the wide range of nanomaterials, nanoparticles are the simplest forms of such unique materials from the point of view of nanostructures. Despite the simplicity, nanoparticle-based technologies broadly cover diverse fields, such as polymer science and technology, energy generations, chemical storages, environmental remediations, microelectronics, electro-optical and optical sciences, medical diagnostics, medical therapy, foods, cosmetics, structural materials, and so on [1,2,3,4,5,6,7]. Nanoparticles can be classified into three general categories of organic nanoparticles, inorganic nanoparticles, and organic–inorganic hybrid nanoparticles. Each category has numerous subcategories. For example, organic nanoparticles are subcategorized into (i) small-molecule-based micelles and vesicles, (ii) polymeric micelles and vesicles, and (iii) polymeric nanoparticles.

The utilization of nanoparticles in various applications requires detailed information regarding their structures and properties. In particular, the structure of a nanoparticle is known to influence the overall properties, and, therefore, is the factor holding greater importance in the evaluation of the nanoparticle’s suitability for applications [7]. Nanoparticle structure as a parameter could be described from a more detailed and meticulous manner through the characterization of nanoparticle shape, morphological composition, surface roughness, surface texture, surface functionality, surface charge, degree of inter-nanoparticle aggregation, size (i.e., surface-to-volume ratio), size distribution, and so on. Amongst the various aspects of nanoparticle structure, the size and size distribution are the most significant features in the perspectives of nanotechnology and engineering, since tailoring desired property of nanoparticles with high precision requires control over the nanoparticle size and size distribution in the fabrication process. Fabricating homogeneous, monodisperse nanoparticles is the most desirable case, but all known synthetic routes producing nanoparticles including post-fabrication processes cannot avoid causing certain degrees of size distribution and structural imperfections. So far, a variety of analysis methods have been applied to determine particle size for the past few decades of nanotechnology research [7,8,9,10,11,12,13,14,15,16,17]. The size analysis methods can be categorized into two families: (i) ensemble techniques and (ii) counting techniques [14]. Ensemble techniques can provide a wide dynamic size range and high statistical accuracy. In comparison, counting techniques give a narrow dynamic size range and low statistical accuracy; counting methods are more suitable to measure a few nanoparticles beyond selected size limits. The ensemble technique family includes dynamic light scattering (DLS, which is so-called photon correlation spectroscopy, diffusing-wave spectroscopy or quasi elastic light scattering), static light scattering, laser diffraction, X-ray scattering, neutron scattering, fluorescence correlation spectroscopy, sedimentation, and sieving [8,9,10,11,12,13,14,15,16,17]. The counting technique family includes transmission electron microscopy, scanning electron microscopy, atomic force microscopy, optical particle counting, electrozone sensing, and resistive pulse sensing (which is also known as Coulter counting) [8,9,10,11,12,13]. 

In the last decades, a great progress has been made in developing analysis schemes for DLS data [13,18,19,20,21,22,23,24,25]. As a result, DLS characterization has become a popular tool, mostly due to the easy measurement and automatized data analysis outputs which are provided by commercial DLS instruments produced in compact size and reasonable price [25,26,27,28,29,30,31]. Specifically, their measurement and data analysis software packages have been developed for spherical particles [25]. Nevertheless, the easy-to-use, compact DLS systems are widely utilized to characterize nanoparticles or macromolecules of various non-spherical shapes. It is, then, crucial to verify the precision and accuracy levels, and the limits of such DLS systems regarding the measured raw data and data analysis results when the subjects of the measurement are non-spherical.

In this study we have, therefore, attempted comparative DLS and synchrotron X-ray scattering analyses on different shaped particles prepared of topological amphiphilic block copolymers. Interestingly, a *cyclic* poly(*n*-decyl glycidyl ether-*block*-2-(2-(2-methoxyethoxy)ethoxy)ethyl glycidyl ether) (*c*-PDGE-*b*-PTEGGE, a *cyclic* amphiphile) and its *linear* analogue (*l*-PDGE-*b*-PTEGGE, a *linear* amphiphile) are known to form oblate and prolate ellipsoidal micelles respectively [32]. Moreover, we have recently synthesized a poly(6-phosphorylcholinehexylthiopropyl glycidyl ether-*block*-*n*-dodecanoyl glycidyl ether) (*l*-PPCGE-*b*-PDDGE) as another *linear* amphiphile having a hydrophilic bristle-based block different from that of *c*-PDGE-*b*-PTEGGE and its linear analogue. The micelles of these three topological amphiphiles have been selected in this study as nanoparticle models in different shapes. This comparative study found that there are large differences in the structural parameters determined by DLS and X-ray scattering. Additional information such as micelle shape and structural details were extracted by quantitative X-ray scattering data analysis. We try to understand such large differences by considering all possible factors and reasons. We propose a better use of commercialized DLS instrument with keeping its merits. 

## 2. Materials and Methods

### 2.1. Materials

A *cyclic* amphiphile, *c*-PDGE-*b*-PTEGGE, and its *linear* analogue (*l*-PDGE-*b*-PTEGGE) were prepared from *n*-decyl glycidyl ether and 2-(2-(2-methoxyethoxy)ethoxy)ethyl glycidyl ether in accordance to the methods reported recently in the literature [33]. In addition, another *linear* amphiphile, *l*-PPCGE-*b*-PDDGE was prepared according to the methods in the literature [34,35]. The chemical structures and material information of the topological block copolymers are given in Figure 1 and Table 1, respectively.

### 2.2. Micelle Formations

*c*-PDGE-*b*-PTEGGE was dissolved first in ethanol and then deionized water was slowly added, producing a micellar solution of 0.5 wt % concentration in a mixture of 75 wt % ethanol and 25 wt % deionized water. In the same manner, a micellar solution of *l*-PDGE-*b*-PTEGGE was prepared. The micellar solution of *l*-PPCGE-*b*-PDDGE was prepared with a concentration of 0.5 wt % in chloroform. All micellar solutions were filtered using disposable syringes equipped with a polytetrafluoroethylene filters with a pore size of 0.2 μm before DLS and X-ray scattering measurements.

### 2.3. Measurements

DLS measurements were conducted at 25 °C using a DLS instrument (model Zetasizer Nano ZS90, Malvern Instruments Ltd., Worcestershire, UK). The detector was positioned at 90°; in other words, the scattering angle was 90°. A He-Ne laser of 632.8 nm wavelength was used. 

Solution X-ray scattering measurements were carried out at the 4C beamline [16,17,37] of the Pohang Accelerator Laboratory. All scattering measurements were carried out with sample-to-detector distances (SDD) of 4 m and 1 m at room temperature. A quartz capillary tube (1.5 mm inner diameter) was used as a solution cell; 50 μL of polymer solution was loaded in the sample cell. The scattering data were collected through a two-dimensional (2D) charge-coupled detector (CCD) (model Rayonix 2D Mar, Evanston, IL, USA) using an X-ray radiation source (λ = 0.0756 nm, wavelength) with an exposure time of 60 s. The scattering angle was calibrated with precalibrated polystyrene-*b*-polyethylene-*b*-polybutadiene-*b*-polystyrene block copolymer and silver behenate powder (Tokyo Chemical Industry (TCI), Tokyo, Japan). The 2D scattering data were circularly averaged with respect to the beam center and normalized to the intensity of the transmitted X-ray beam monitored via a scintilliation counter positioned behind the sample. The scattering data were further corrected for the scattering due to the solvent.

### 2.4. Dynamic Light Scattering (DLS) Data Analysis

The particles dispersed homogeneously in a solution scatter light in all directions (which is so-called Rayleigh scattering) upon exposure. The scattered light intensity is always fluctuating over time, which is attributed to continuous changings in the interdistances of particles that undergo Brownian motion in the solution. This scattered light then undergoes either constructive or destructive interference by the surrounding particles to produce intensity fluctuation that is unique to the particles regarding the time scale of movement of the particles. The dynamic information of the particles is derived from an autocorrelation of the scattered intensity trace recorded during the experiment. The autocorrelation curve (i.e., second order autocorrelation function *g*_2_(*q,t*)) is generated from the fluctuating scattered intensity *I*_s_*(q,t)* trace as follows [18,19,20]:(1)$$g_{2}(q,\tau )=\frac{\langle I_{s}(q,t)I(q,t+\tau )\rangle }{\langle {\left|I_{s}(q,t)\right|}^{2}\rangle }$$
with *t* is the time, *τ* is the time delay. *q* is the magnitude of scattering vector which is defined by
(2)$$q=\frac{4\pi n_{o}}{\lambda }\sin\theta$$
where *n*_o_ is the refractive index of a solvent used, λ is the wavelength of a laser light used, and 2*θ* is the scattering angle. By means of the Siegert relation [38,39], *g*_2_(*q*,*τ*) can be related to the field correlation function *g*_1_(*q*,*τ*) (i.e., first order correlation function) [18,19,20]:(3)$$g_{2}(q,\tau )=1+\beta {\left|g_{1}(q,\tau )\right|}^{2}$$
with *β* is a correction factor that depends on the geometry and alignment of the laser beam in the light scattering setup. 

For the particles in a monodisperse population, *g*_1_(*q*,*τ*) can be expressed as a single exponential decay as follows:(4)$$g_{1}(q,\tau )=\exp(-\Gamma \tau )$$
where *Γ* is the decay rate constant. Here, *Γ* is related to the translational diffusion coefficient *D_t_* (i.e., intensity weighted diffusion coefficient) of the particles (i.e., scatterers) at a single *q* or a range of *q*:(5)$$\Gamma =q^{2}D_{t}$$

From *D_t_*, the hydrodynamic radius *R_h_* of the particle (specially, *spherical hard particle*) can be calculated by using the Stokes–Einstein equation [40,41] and the viscosity of the surrounding medium:(6)$$R_{h}=\frac{k_{B}T}{6\pi \eta D_{t}}$$
where *k_B_* is the Boltzmann constant (1.380 × 10^−23^ kg·m^2^/s^2^·K), and *T* and *η* are the absolute temperature and viscosity of the solution medium respectively. 

For the particles in a polydisperse population, *g*_1_(*q*,*τ*) can be described as an intensity weighed integral over a distribution of decay rates *G_i_*(*Γ**_i_*) (i.e., *G*(*Γ*)) corresponding to each of the species in the population:(7)$$g_{1}(q,\tau )=\sum_{i=1}^{n}G_{i}(\Gamma_{i})\exp(-\Gamma_{i}\tau )=\int G(\Gamma )\exp(-\Gamma \tau )d\Gamma$$

To analyze *g*_2_(*q*,*τ*) data, the Cumulant method [13,21,22] has been developed and widely employed. This analysis method is currently adopted for all DLS instruments commercialized so far. Here, it is noted that the Cumulant method is valid for small *τ* and sufficiently narrow *G*(*Γ*); this method is only suitable for a Gaussian-like distribution around the mean values. Thus, a research effort has been made to develop more appropriate data analysis methods. A representative of the improved methods is the CONTIN algorithm [23,24], which is currently adopted widely to analyze *g*_2_(*q*,*τ*) data of particles with a polydispersity. Based on the Taylor expansion, the *g*_1_(*q,**τ*) of polydisperse particles can be rewritten as:(8)$$g_{1}(q,\tau )=\exp\{-\bar{\Gamma }(\tau +\frac{k_{2}}{2!}\tau^{2}-\frac{k_{3}}{3!}\tau^{3}+\frac{k_{4}}{4!}\tau^{4}\ldots \ldots \ldots )\}$$
where *k*_2_, *k*_3_, and *k*_4_ represent the variance, skewness, and kurtosis of measured distributions respectively for the decay rates of Gaussian distribution. $\bar{\Gamma }$ is the mean of decay rate constant *Γ* and can be expressed as the following equation:(9)$$\bar{\Gamma }=\int G(\Gamma )\Gamma d\Gamma =k_{1}(\tau ).$$
$\bar{\Gamma }$ is related to the *z*-averaged translational diffusion coefficient *D_z_* (i.e., *z*-averaged intensity weighted diffusion coefficient) of the particles:(10)$$\bar{\Gamma }=q^{2}D_{z}$$

From *D_z_*, the *z*-averaged hydrodynamic radius *R_h,z_* of the particle (specially, *spherical particle*) can be calculated as:(11)$$R_{h,z}=\frac{k_{B}T}{6\pi \eta D_{z}}$$

From $\bar{\Gamma }$ and *k*_2_, the second order polydispersity index *PDI_DLS_* (i.e., DLS polydispersity index: an indication of the variance) of particles can be further derived by using the following relation:(12)$$PDI_{DLS}=\frac{k_{2}}{{\bar{\Gamma }}^{2}}.$$

### 2.5. X-ray Scattering Data Analysis

The scattering intensity *I*(*q*) of particles in a solvent medium is expressed by the following equation [42]:(13)$$I(q)=K_{x}N_{p}P(q)\cdot S(q)$$
where *K_x_* is a constant, *N_p_* is the number of particles, *P*(*q*) is the form factor of the particle, and *S*(*q*) is the structure factor for the particles. *q* is the magnitude of the scattering vector defined as *q* = (4π/λ)sin*θ* in which 2*θ* is the scattering angle and λ is the wavelength of the X-ray beam used. The *P*(*q*) term contains the information of the shape and size of the particle while the *S*(*q*) term describes the interparticle distance as the following [41,42]:(14)$$S(q)=1+\frac{1}{N}\langle \sum_{l=1}^{N}\sum_{l’\neq 1}^{N}e^{-jq(r_{l}-r_{l’})}\rangle$$
where *r_l_* is the position of particle *l*. The first term of the structure factor equals 1 due to the perfect positional correlation stemming from particle *l* itself. The second term is the interference function between particles. 

For general solution scattering cases in dilute conditions, the distances between particles (*r_l_ – r_l’_*) become random and the interference function reaches 0:(15)$$\frac{1}{N}\langle \sum_{l=1}^{N}\sum_{l’\neq 1}^{N}e^{-jq(r_{l}-r_{l’})}\rangle \to 0$$

Then,
(16)S(q)≈1
thereby allowing the approximation of the *S*(*q*) term as unity over the entire *q* range. This approximation allows the isolation of *P*(*q*) from the overall scattering intensity *I*(*q*). Therefore, Equation 13 can be rewritten as:(17)$$I(q)≅K_{x}N_{p}P(q)$$

Dilute conditions, however, present the problem of having considerably low intensity in the high *q* region as opposed to the low *q* region, which is detected in much higher intensity. This problem causes the intensity in the high *q* region to be obscured by the background noise, producing significant errors in analysis of the scattering data. As a way of overcoming such low intensity, semidilute conditions are often used with the assumption that the interparticle interaction is low enough where *S*(*q*) ≈ 1 over the entire *q* range. In this study, the X-ray scattering measurements were conducted with 0.5 wt % of polymeric micelles in solution to obtain high quality scattering data with negligible *S*(*q*).

#### 2.5.1. Guinier Analysis 

The Guinier analysis is a model independent method based on the law of Guinier, which is expressed as the following equation [42]:(18)$$\ln I(q)=\ln I_{o}\left(q\right)-\frac{q^{2}R_{g,G}{}^{2}}{3}$$
where *I*_o_(*q*) is the incident beam intensity. According to the law of Guinier, the radius of gyration *R_g,G_* of particles in solution can be estimated from the low *q* region of the scattering data. This determination, however, must satisfy the two boundary conditions: the particles must be globular and the maximum *qR_g,G_* must be less than 1.33. Within these boundary conditions, the obtained scattering data can be plotted into a ln*I*(*q*) vs. *q*^2^ curve, in which the resulting slope is used to yield *R_g,G_*.

#### 2.5.2. Indirect Fourier Transformation (IFT) Analysis 

The indirect Fourier transformation (IFT) analysis is another model independent method, which can give structural information. The radius of gyration *R_g_* can also be obtained through the IFT method [43,44], which transforms the scattering data to its real space analogue, pair distance distribution function *p*(*r*). The pair distance distribution function describes the probability of finding two scatterers separated by a distance *r* inside the particle (i.e., micelle). This data transformation bypasses the utilization of a specific parameterized model and directly derives the structural information from the scattering data. The scattering intensity can be expressed by the Fourier transformation of *p*(*r*) [43,44]:(19)$$I(q)=4\pi \int_{0}^{\infty }p(r)\frac{\sin(qr)}{qr}dr$$
in which an estimate of *R_g,IFT_* can be derived based on *p*(*r*) as follows:(20)$$R_{g,IFT}=\frac{\int_{0}^{\infty }p(r)r^{2}dr}{2\int_{0}^{\infty }p(r)dr}.$$

#### 2.5.3. Three-Phase Ellipsoid Model Analysis

The three-phase ellipsoid model is composed of three phases: (i) a dense core, (ii) a dense corona, and (iii) a solvated corona [32]. For this three-layer ellipsoid particle, the *P*(*q*) in Equation (17) can be expressed by the sum of two terms in the following equation:(21)$$P(q)=P_{shape}(q)+P_{blob}(q)$$
where the first term *P_shape_*(*q*) describes an ellipsoidal particle consisting of three phases, namely a core, a dense corona, and a solvated corona. The core is unpenetrated by solvent molecules and the corona is divided into two regions of different densities depending on the level of solvent penetration. The second term *P_blob_*(*q*) describes the density fluctuations smaller than the blob radius within the corona regions. 

In *P_shape_*(*q*), the form factor describes a spheroid shape with a pair of equal semi-axes (*R*, *R*) and a distinct third semi-axis (*εR*) (i.e., an ellipsoid of gyration). The ellipsoidicity ratio *ε* in this model is defined as the ratio between the polar axis (*R_p_*) and the two equatorial axes (*R_e_*):(22)$$\varepsilon =\frac{R_{p}}{R_{e}}$$

For a single spheroid with the equatorial radius of *R* (i.e., *R_e_*), the form factor is described as [45,46]:(23)$$P_{shape}(q)=\int_{0}^{\frac{\pi }{2}}F^{2}[q,r(R_{e},\varepsilon ,\varphi )]\sin\varphi \;d\varphi$$
where *F*[*q*,*r*(*R_e_*,*ε*,*φ*)] is the scattering amplitude and *φ* is the modular angle of spheroid. The scattering amplitude is given as a function of the scattering vector *q*, and *r*(*R_e_*,*ε*,*φ*):(24)$$F[q,r(R_{e},\varepsilon ,\varphi )]=\frac{3\{\sin[qr(R_{e},\varepsilon ,\varphi )]-qr(R_{e},\varepsilon ,\varphi )\cos[qr(R_{e},\varepsilon ,\varphi )]\}}{{\left[qr(R_{e},\varepsilon ,\varphi )\right]}^{3}}$$
where,
(25)$$r(R_{e},\varepsilon ,\varphi )=R_{e}{\left(\sin^{2}\varphi +\varepsilon^{2}\cos^{2}\varphi \right)}^{\frac{1}{2}}$$

In the case of the three-phase ellipsoid model, the overall form factor can be expressed thus:(26)$$\begin{array}{ll} P_{shape}(q)= & \int_{0}^{\frac{\pi }{2}}\{(\rho_{s.corona}V_{s.corona}F_{s.corona}^{2})+[(\rho_{s.corona}-\rho_{d.corona})V_{d.corona}F_{d.corona}^{2}]\\ & +[(\rho_{d.corona}-\rho_{core})V_{core}F_{core}^{2}]\}\sin\varphi d\varphi . \end{array}$$

Additionally, each scattering amplitude term includes an interface term that functions to describe the gradual decay of polymeric micelle’ density along the radial direction. Because polymeric micelles are soft particles, they do not possess sharp interface between phases, and the region of fuzzy, nonideal interface is especially prevalent in the dense and solvated corona. The resulting scattering amplitude can be rewritten as:(27)$$F[q,r(R_{e},\varepsilon ,\varphi ),t_{f}]=F[q,r(R_{e},\varepsilon ,\varphi )]\cdot e^{-\frac{q^{2}t_{f}^{2}}{4}}$$
where 2*t_f_* is the width (i.e., thickness) of the fuzzy interface. Moreover, the core radius, dense corona thickness, and solvated corona thickness within *P_shape_*(*q*) are assumed to possess Gaussian distribution *n*(*A*):(28)$$n(A)=\frac{1}{\sqrt{2\pi }\sigma_{A}}\cdot e^{-\frac{{\left(A-\bar{A}\right)}^{2}}{2\sigma_{A}^{2}}}$$
where *A* corresponds to the core radius or the dense corona thickness or the solvated corona thickness, $\bar{A}$ is the mean value of the structural parameter *A*, and *σ_A_* is the standard deviation of *A* from $\bar{A}$. Therefore, Equation (27) can be rewritten as:(29)$$F[q,r(R_{e},\varepsilon ,\varphi ),t_{f},\sigma_{A}]=n(A)F[q,r(R_{e},\varepsilon ,\varphi )]\cdot e^{-\frac{q^{2}t_{f}^{2}}{4}}$$

The second term of the overall form factor, *P_blob_*(*q*), is expressed as the following [47]:(30)$$P_{blob}(q)=4\pi \alpha \int_{0}^{\xi }r^{2}\gamma (r)\cdot \frac{\sin(qr)}{qr}dr$$

Here, *P_blob_*(*q*) is the contribution from the density fluctuations in the corona shell. It describes the Fourier transform of the correlation function *γ*(*r*) of density fluctuations on length scales (*r*) smaller than the blob radius, *α* is the amplitude of the blob scattering contribution, and *ξ* is the average correlation length. *γ*(*r*) is equal to zero for *r* > *ξ*, but not equal to zero for *r* ≤ *ξ*. For *r* ≤ *ξ*, *γ*(*r*) can be expressed by:(31)$$\gamma (r)\propto r^{\mu -2}$$
where,
(32)$$\mu =\chi^{-1}-1$$
*χ* is the Flory–Huggins parameter, which equals 3/5 for the good solvent condition, 1/2 for the Θ solvent condition, and 2/3 in the case that the molecules are stretched [15,32,47]. 

#### 2.5.4. Three-Phase Cylinder Model Analysis

This model is identical to the three-phase ellipsoid model regarding the composition of a dense core, a dense corona, and a solvated corona, and the consideration of blob scattering. The shape of the model, however, is cylindrical. The equation for a single right circular cylinder could be written as: (33)$$P_{shape}(q)=\int_{0}^{\frac{\pi }{2}}{\left[\frac{2B_{1}(qR\sin\alpha )}{qR\sin\alpha }\cdot \frac{\sin\{(qL\cos\alpha )/2\}}{(qL\cos\alpha )/2}\right]}^{2}\sin\varphi \;d\varphi$$
where *B*_1_(*x*) is the first order Bessel function, *R* is the radius of the cylinder, *L* is the height of the cylinder, and *φ* is the modular angle [45]. The expression for a cylinder consisting of three phases can then be constructed in the same approach used in the three-phase ellipsoid model analysis (Equation (26)). Equations (27) and (27) are also implemented into this model to account for the region of fuzzy, non-ideal interfaces between all the phases and Gaussian distribution of the dimensions of each phase. Finally, the blob scattering term *P_blob_*(*q*) from Equation 30 can be combined with the overall three phase cylinder form factor term *P_shape_*(*q*) through summation as shown in Equation (21).

## 3. Results and Discussion

In this study, we have tried to investigate micellar solutions of a cyclic diblock amphiphile (*c*-PDGE-*b*-PTEGGE) and two linear diblock amphiphiles (*l*-PDGE-*b*-PTEGGE and *l*-PPCGE-*b*-PDDGE) using DLS and X-ray scattering analyses.

Figure 2a shows a representative of the autocorrelation function *g*_1_(*q*,*τ*) data measured for the micelles of *c*-PDGE-*b*-PTEGGE in an ethanol/water (75/25 in *wt/wt*) mixture by using DLS analysis. As presented in Figure 2b, the *g*_1_(*q*,*τ*) data are reasonably well fitted by using the Zetasizer program of Malvern Instruments which was developed with the Cumulant method [21,22,25] and CONTIN algorithm [23,24,25]. In this analysis, the DLS polydispersity index (*PDI_DLS_*) is found to be 0.03. This *PDI_DLS_* value is much smaller than 0.70, which is the upper limit for good quality DLS data analysis [13]. Therefore, the obtained *PDI_DLS_* value indicates that the data analysis was done well; the very low *PDI_DLS_* value further suggests that the micelles were formed in narrow distribution (i.e., nearly monodisperse distribution). This is confirmed in the obtained intensity-, number-, and volume-weighted distributions that show a unimodal peak, respectively (Figure 2c–h). The analysis results are summarized in Table 2. The hydrodynamic radius is determined to be 21 nm (= *R*_*h*,1*i*_) from the intensity-weighted distribution, 17 nm (= *R*_*h*,1*n*_) from the number-weighted distribution, and 19 nm (= *R*_*h*,1*v*_) from the volume-weighted distribution. The *z*-averaged hydrodynamic radius *R_h,z_* is 20 nm.

The micelle solution was further subjected to X-ray scattering analysis. Figure 3a is a representative of the measured X-ray scattering data. The scattering data were analyzed in a comprehensive manner by using the Guinier law, IFT method, and three-phase ellipsoidal model approach described above. The pair distance distribution function *p*(*r*) is found to reveal a bell shape profile with very little distortion (Figure 3b), informing that the *c*-PDGE-*b*-PTEGGE micelle is formed in a slightly distorted sphere, i.e., ellipsoidal shape. The scattering profile could be successfully well fitted by the three-phase ellipsoidal model approach (Figure 3c), confirming the formation of ellipsoidal micelle. The micelle is determined to have an ellipsoidal structure consisting of core (6.10 nm radius), dense corona (3.85 nm thick), and soft corona (1.60 nm thick). The analysis results are presented in Figure 3d,e; the obtained structural parameters are summarized in Table 2. The radius of gyration of the micelle is 7.75 nm (= *R_g,G_*) from the Guinier analysis, 7.97 nm (= *R_g,IFT_*) from the IFT analysis, and 8.50 nm (= *R_g,TPS_*) from the three phase ellipsoidal (which is a three phase structure (TPS)) model approach. *R_g,G_*/*R_g,IFT_* = 0.97, which is slightly lower than 1 (for spherical shape). *R_max_*/*R_g,IFT_* = 1.33, which is slightly lower than 1.36 (for spherical shape); here, *R_max_* is the radius of micelle determined from the peak maximum of the *p*(*r*) function profile. *D_max_*/*R_max_* = 2.17, which is larger than 2 (for sphere); *D_max_* is the maximum dimension (i.e., diameter) of micelle determined from the *p*(*r*) function. The ellipsoidicity (*ε_el_* = polar radius (*R_p_*)/equatorial radius (*R_e_*)) is 0.84, which is deviated from 1 (for sphere). These results collectively indicate that the *c*-PDGE-*b*-PTEGGE micelle is an oblate ellipsoid in shape, which is different from a sphere assumed in the DLS analysis.

All radius values of the micelle, which are determined by X-ray scattering analyses, are 1.5 to 2.2 times smaller than the hydrodynamic radii (*R*_*h*,1*i*_, *R*_*h*,1*n*_, *R*_*h*,1*v*_, and *R_h,z_*) measured by DLS analysis. The radii of gyration are 2.0 to 2.7 times smaller than the hydrodynamic radii (*R*_*h*,1*i*_, *R*_*h*,1*n*_, *R*_*h*,1*v*_, and *R_h,z_*). Instead, the hydrodynamic radii are more close to the maximum dimension (*D_max_*, i.e., diameter) of micelle. Moreover, the radius distribution, which was obtained by DLS analysis, is much broader than that determined by X-ray scattering. Overall, there are large differences in the micelle sizes and distributions measured by DLS and X-ray scattering analyses. 

A representative DLS profile of the *l*-PDGE-*b*-PTEGGE micelles in an ethanol/water (75/25 in *wt/wt*) mixture is presented in Figure 4a. The DLS data are analyzed, as shown in Figure 4b. In the data analysis, *PDI_DLS_* = 1.00, which is larger than 0.70. It is generally known that *PDI_DLS_* > 0.70 is an indication of broad particle size distribution, namely multimodal particle size distribution [13]. From this analysis, a trimodal radius distribution is determined for the micelle (Figure 4c–e). The structural parameters obtained are listed in Table 3. The first group (Peak 1) as a major micelle component is found to have a hydrodynamic radius *R_g_/R*_*h*,1_ of 62 to 78 nm depending on the intensity-, number-, and volume-weighted distributions. The second group (Peak 2) as a minor micelle component has a hydrodynamic radius *R*_*h*,2_ of 430 to 488 nm. The third group (Peak 3) has a hydrodynamic radius *R*_*h*,3_ of 2780 to 2795 nm but is negligible because it could not be discernible in the number-weighted distribution. For this multimodal distribution, *R_h,z_* = 1003 nm. This value seems to be overestimated due to the inclusion of the third micelle group in the largest size which possibly is not present in the solution. Thus, the *R_h,z_* value covering the first and second micelle groups should be in a hundred nanometer or less, rather than such the high value (1003 nm). The results collectively inform that *l*-PDGE-*b*-PTEGGE forms micelles with very large size and broad multimodal distribution, which are quite different from its cyclic analogue micelle. Overall, the DLS analysis results are quite unusual. 

Figure 5a shows a representative of the X-ray scattering data measured for the *l*-PDGE-*b*-PTEGGE micelle solution. The scattering data were analyzed in a quantitative manner as carried out for the micelle of the cyclic amphiphile analogue. The analysis results are presented in Figure 5b–e and Table 3. The *p*(*r*) function profile is clearly asymmetric (Figure 5b), which is far from a symmetrical bell shape being observed for the sphere. For this non-spherical nature, more clues are found in structural parameters such as *R_g,G_*/*R_g,IFT_*, *R_max_*/*R_g,IFT_*, and *D_max_*/*R_max_*. The scattering data could be satisfactorily fitted by using the three phase ellipsoidal approach, as shown in Figure 5c. This analysis finds that *l*-PDGE-*b*-PTEGGE forms an ellipsoidal micelle which is structurally composed of core (23.00 nm radius), dense corona (6.80 nm thick), and soft corona (4.60 nm thick). Moreover, it is confirmed that *l*-PDGE-*b*-PTEGGE makes micelles with a unimodal distribution as observed for the cyclic analogue. However, the *l*-PDGE-*b*-PTEGGE micelle is a prolate ellipsoid rather than an oblate ellipsoid. The prolate ellipsoid is determined to have ε*_el_* = 1.50, informing that the level of distortion from sphere is much higher in comparison to that of the micelle of the cyclic amphiphile analogue. The radius of gyration of the *l*-PDGE-*b*-PTEGGE micelle is in a range of 18.05−22.20 nm. The micelle has a radius in a range of 22.20−34.50 nm. These micelle parameters are relatively much larger than those of the micelle of the cyclic amphiphile analogue. The radius distribution is relatively broader than that of the cyclic amphiphile micelle.

In the views of micelle size and size distribution, surprisingly there are huge mismatches between the results of DLS and X-ray scattering analyses. The hydrodynamic radius, which is determined by DLS analysis, ranges in 62 to 488 nm. This broad range of hydrodynamic radius is 2 to 22 times larger than that (22.20−34.50 nm) measured by X-ray scattering analysis. The hydrodynamic radius is also much larger than the radius of gyration measured by X-ray scattering. 

Figure 6a shows a representative of the DLS data measured for the *l*-PPCGE-*b*-PDDGE micelle solution in chloroform. The DLS data could be reasonably well analyzed (Figure 6b), giving *PDI_DLS_* = 0.22. Even though the low *PDI_DLS_* value, bimodal peaks are observed in both the intensity- and volume-weighted distributions, a unimodal peak is observed in the number-weighted distribution (Figure 6c−e). The analysis results are summarized in Table 4. The hydrodynamic radius ranges in 7−42 nm, depending on the distribution peaks and weighted distributions; *R_h,z_* = 32 nm. 

Figure 7a presents the X-ray scattering data of the micelle solution. The IFT analysis of the scattering gives an asymmetric *p*(*r*) function profile, as shown in Figure 7b. Taking into consideration the *p*(*r*) function profile, we have tried to analyze the scattering data by using the three-phase ellipsoidal approach but failed. As a result of the data analysis efforts with various structural models, it is found that the scattering data could be satisfactorily fitted by using the three-phase cylindrical model approach (Figure 7c). The micelle is determined to have a cylindrical structure that consists of core (2.00 nm radius and 1.50 nm height), dense corona (2.30 nm thick along the short axis and 6.60 nm thick along the long axis), and soft corona (3.40 nm thick along the short axis and 3.80 nm thick along the long axis). The micelle is formed in a very narrow unimodal size distribution (Figure 7d−g). The determined structural parameters are listed in Table 4. The radius of gyration of the cylindrical micelle ranges in 8.43−8.61 nm. The short axial length of the micelle is in the range of 7.70−9.95 nm (*R_cyl_* and *R_max_*); the long axial length ranges in 22.30−27.30 nm (*H_cyl_* and *D_max_*). *R_g,G_/R_g,IFT_* = 0.98; *R_max_/R_g,IFT_* = 1.16; *D_max_/R_max_* = 2.74. These results strongly support a cylindrical structure of the micelle. Overall, the X-ray scattering analysis results of *l*-PPCGE-*b*-PDDGE micelle are quite different from those of the DLS analysis.

For all micelle solutions of this study, there are large differences in the sizes and size distributions determined by DLS and X-ray scattering analyses, as described above. These differences could be caused in several ways as follows.

First, DLS analysis actually determines the hydrodynamic radius *R_h,z_* of a particle including not only the particle itself but also any possible solvent layers associated with it in solution. Thus, one expects that DLS analysis may provide relatively larger particle size value, compared to that determined by X-ray scattering analysis.

Second, the DLS data analysis based on the Cumulant method and CONTIN algorithm is generally performed under an important assumption that micelles in solution have spherical shapes. However, the quantitative X-ray scattering analyses confirmed an oblate ellipsoid structure with ε*_el_*= 0.84 (ellipsoidicity) for the normal *c*-PDGE-*b*-PTEGGE micelle, a prolate ellipsoid structure with ε*_el_*= 1.50 for the normal *l*-PDGE-*b*-PTEGGE micelle, and a cylindrical structure with ε*_cyl_* = 1.45 (aspect ratio) for the reverse *l*-PPCGE-*b*-PDDGE micelle. Therefore, the assumption of spherical micelle in the DLS data analysis might cause significant errors in the size and size distribution outputs for the non-spherical micelles of this study.

Third, the X-ray scattering analyses found that the individual micelle solutions have narrow unimodal size distributions. However, the DLS analyses gave a broad unimodal size distribution for the oblate ellipsoid micelles of *c*-PDGE-*b*-PTEGGE in which the level of distortion from sphere is relatively low (ε*_el_* = 0.84) but broad multimodal size distributions for the prolate ellipsoid and cylindrical micelles of the other two polymers in which the levels of distortion from sphere are relatively large (ε*_el_* = 1.50 and ε*_cyl_* = 1.45). From these comparisons, it is suspected that higher distortion from sphere in micelle shape causes significant errors on the micelle size distribution and, thus, leads large error on the micelle size measured by DLS. In fact, it was previously reported that DLS intensity is primarily dependent on the translational motion of particle and additionally affected by its rotational motion in which the particle does Brownian motion in solution; therefore, for a non-spherical particle, the contribution of its rotational motion cannot be ignored in the DLS data analysis [48,49]. In the field of DLS and its applications with the DLS instruments commercialized with an easy hand-on concept, it is generally used that the hydrodynamic radius *R_h,z_* of a non-spherical particle is the radius of a sphere that has the same translational diffusion speed as the particle. However, such the hydrodynamic equivalent spherical radius may be valid under the condition that DLS analysis is conducted in a proper, quantitative way including both the translational and rotational motions of particles in solution. Otherwise, the *R_h,z_* value and radius distributions include significant errors.

Fourth, the radius of gyration *R_g_* is defined as an average root mean squared distance from the center of the mass of a particle. Thus, the ratio of *R_g_* to *R_h,z_* is often used to get information about the shape of the particle; in the field of biomacromolecular science, it is known that the ratio is ca. 0.70 for spherical shape, >0.70 for elongated shape, and <0.70 for disk shape [13]. This guideline was applied to the micelle systems of this study; furthermore, this guideline was extended for the hydrodynamic radii determined by DLS analysis. The results are listed in Table 2, Table 3 and Table 4. *R_g_/R_h,z_* is 0.42 for the *c*-PDGE-*b*-PTEGGE micelle, 0.02 for the *l*-PDGE-*b*-PTEGGE micelle, and 0.26 for *l*-PPCGE-*b*-PDDGE micelle. These ratios suggest that all micelles in this study have disk shapes; but these are not matched with those found by X-ray scattering analysis. *R_g_/R_h,1_* is 0.02−0.42 for the *c*-PDGE-*b*-PTEGGE micelle, 0.29−0.36 for the *l*-PDGE-*b*-PTEGGE micelle, and 1.07−1.30 for *l*-PPCGE-*b*-PDDGE micelle; here, *R_g_/R*_*h*,1_ is the averaged hydrodynamic radius of micelles in the highest population. When the guideline is used with *R_g_/R*_*h*,1_, the suggested disk shapes are not matched to those of the *c*-PDGE-*b*-PTEGGE and *l*-PDGE-*b*-PTEGGE micelles. However, the suggested elongated shape may be related to that of the *l*-PPCGE-*b*-PDDGE micelle; this still is in conflict with that suggested by the *R_g_/R_h,z_* guideline. Overall, the *R_g_/R_h,z_* guideline, including the *R_g_/R_h,1_* guideline could not be applicable for the micelles in this study.

Lastly, the measurement mechanism and analytic resolution of DLS analysis are different from those of X-ray scattering analysis. These differences could be reflected in the analysis outputs.

As discussed in Introduction section, the current DLS instruments, which were commercialized in a compact, easy, hand-on base, have been developed with data analysis optimization for spherical particles. Therefore, they may be applicable for characterizing such *spherical particle* solutions. However, due to the very limited capability of data analysis, they cannot be applicable to get structural information with high accuracy and precision for non-spherical particles even in very narrow unimodal distribution as well as for particles, including spherical particles, in multimodal distributions. Otherwise, DLS measurements should be performed in a comprehensive manner and then followed by quantitative data analysis in an off-line base rather than using the data analysis software package built into the instruments.

## 4. Conclusions

The micelle solutions of a *cyclic* polymer amphiphile (*c*-PDGE-*b*-PTEGGE) and two *linear* polymer amphiphiles (*l*-PDGE-*b*-PTEGGE and *l*-PPCGE-*b*-PDDGE) were investigated by DLS and synchrotron X-ray scattering; these two analytic methods were employed in a complementary manner as well as in a comparative manner.

The DLS analysis delivered very limited information, only size and size distribution in very low precision and accuracy of each micelle solution system via the data analysis using a program package built into the DLS instrument. In contrast, the quantitative X-ray scattering analysis provided more structural features such as shape and structural parameter details, in addition to the size and size distribution with high precision and accuracy. 

Moreover, this comparative study found that there are large differences even in the micelle sizes and size distributions obtained from the DLS and X-ray scattering analyses. To understand such large differences, a number of possible factors have been discussed. A most concerning factor is the qualitative DLS data analysis using the data analysis software package developed under assuming only spherical-shaped particles. Such a data analysis software package is applicable for only spherical particles including micelles in narrow distributions. The analysis software package is not applicable for non-spherical particles as well as particles in multimodal distributions; otherwise, the obtained parameters include huge errors, giving incorrect information on the size and distribution. Thus, quantitative DLS measurement and data analysis are always necessary to obtain particle size and distribution information with a high accuracy.

## Figures and Tables

**Figure 1 polymers-10-01347-f001:**
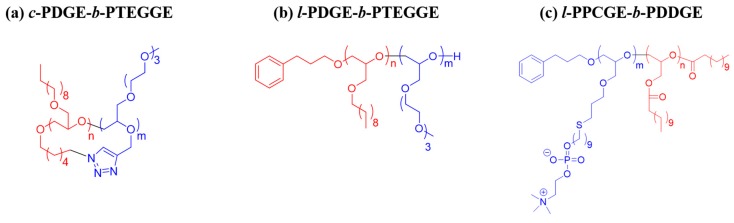
Chemical structures of the topological diblock copolymers in this study.

**Figure 2 polymers-10-01347-f002:**
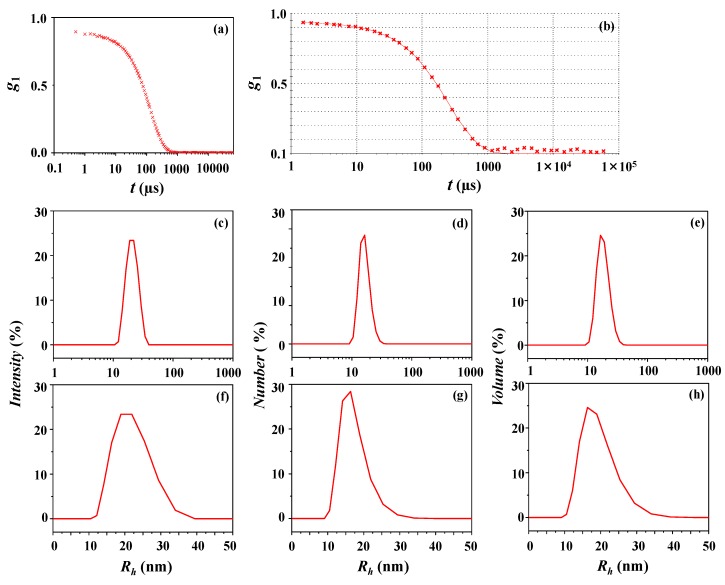
Dynamic light scattering (DLS) data analysis of the *c*-PDGE-*b*-PTEGGE micelle formed in an ethanol/water (75/25 in *wt/wt*) mixture: (**a**) autocorrelation function profile measured at 25 °C; (**b**) data analysis result, where the symbols are the measured data and the red solid line was obtained by fitting the data using the Zetasizer program; (**c**,**f**) intensity-weighted radius distribution obtained by the data analysis; (**d**,**g**) number-weighted radius distribution; (**e**,**h**) volume-weighted radius distribution.

**Figure 3 polymers-10-01347-f003:**
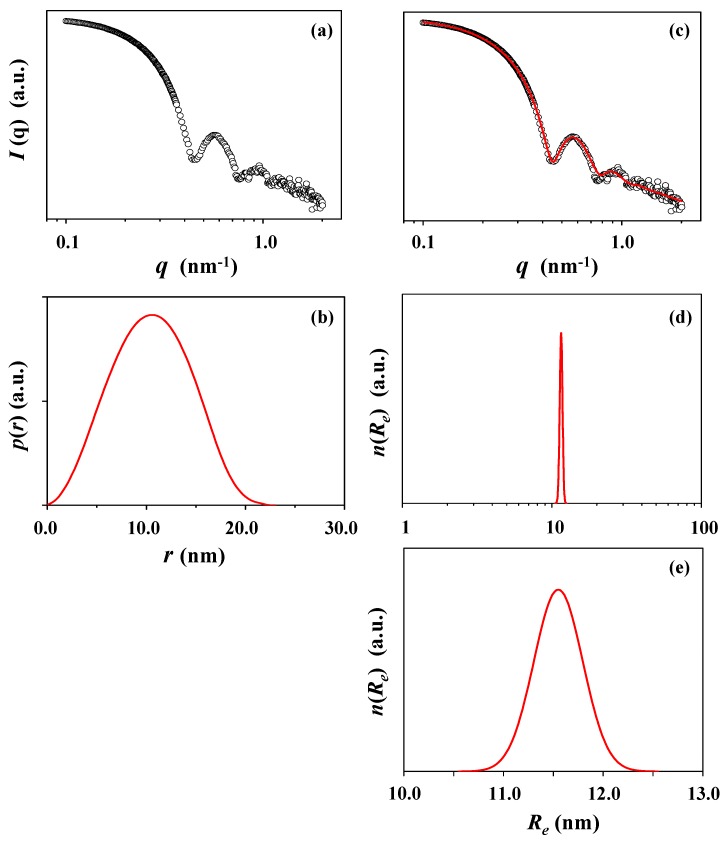
X-ray scattering data analysis of the *c*-PDGE-*b*-PTEGGE micelle formed in an ethanol/water (75/25 in *wt/wt*) mixture: (**a**) scattering data measured at room temperature; (**b**) pair distance distribution functions *p*(*r*) obtained by the data analysis using the IFT method; (**c**) data analysis result, where the open dot symbols are the measured data and the red solid line was obtained by fitting the data using the three phase ellipsoid approach; (**d,e**) radius distribution obtained by the data analysis with the three phase ellipsoid approach.

**Figure 4 polymers-10-01347-f004:**
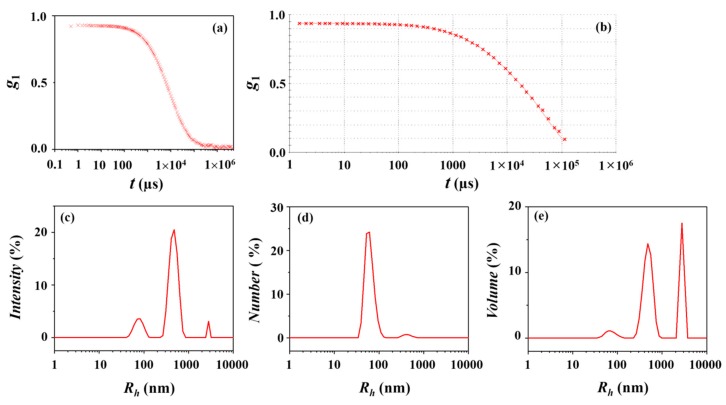
DLS data analysis of the *l*-PDGE-*b*-PTEGGE micelle formed in an ethanol/water (75/25 in *wt/wt*) mixture: (**a**) autocorrelation function profile measured at 25 °C; (**b**) data analysis result, where the symbols are the measured data and the red solid line was obtained by fitting the data using the Zetasizer program; (**c**) intensity-weighted radius distribution obtained by the data analysis; (**d**) number-weighted radius distribution; (**e**) volume-weighted radius distribution.

**Figure 5 polymers-10-01347-f005:**
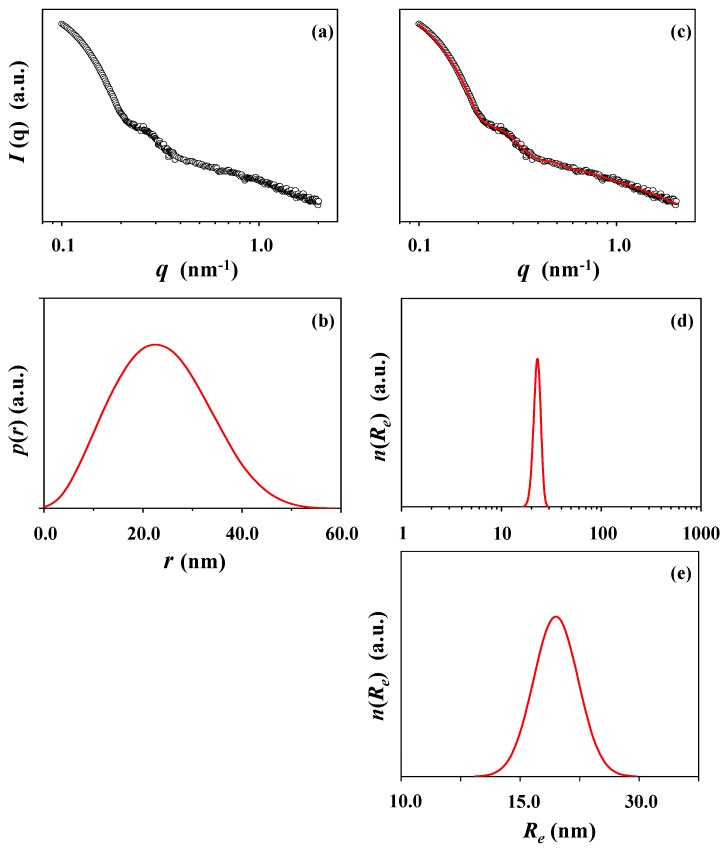
X-ray scattering data analysis of the *l*-PDGE-*b*-PTEGGE micelle formed in an ethanol/water (75/25 in *wt/wt*) mixture: (**a**) scattering data measured at room temperature; (**b**) pair distance distribution functions *p*(*r*) obtained by the data analysis using the IFT method; (**c**) data analysis result, where the open dot symbols are the measured data and the red solid line was obtained by fitting the data using the three phase ellipsoid approach; (**d,e**) radius distribution obtained by the data analysis with the three-phase ellipsoid approach.

**Figure 6 polymers-10-01347-f006:**
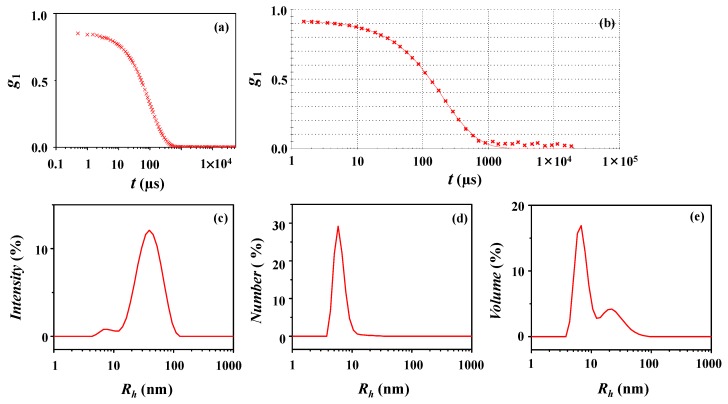
DLS data analysis of the *l*-PPCGE-*b*-PDDGE micelle formed in chloroform: (**a**) autocorrelation function profile measured at 25 °C; (**b**) data analysis result, where the symbols are the measured data and the red solid line was obtained by fitting the data using the Zetasizer program; (**c**) intensity-weighted radius distribution obtained by the data analysis; (**d**) number-weighted radius distribution; (**e**) volume-weighted radius distribution.

**Figure 7 polymers-10-01347-f007:**
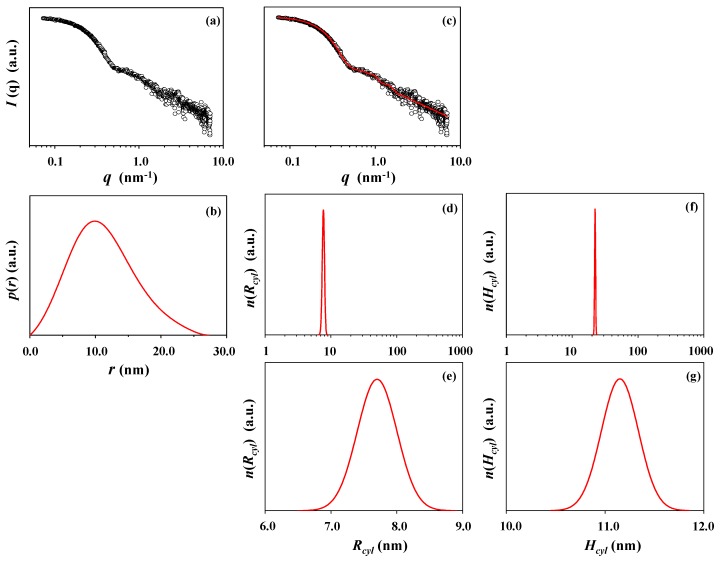
X-ray scattering data analysis of the *l*-PPCGE-*b*-PDDGE micelle formed in chloroform: (**a**) scattering data measured at room temperature; (**b**) pair distance distribution functions *p*(*r*) obtained by the data analysis using the IFT method; (**c**) data analysis result, where the open dot symbols are the measured data and the red solid line was obtained by fitting the data using the three-phase cylinder approach; (**d,e**) radius distribution obtained by the data analysis; (**f,g**) height distribution.

**Table 1 polymers-10-01347-t001:** Molecular characteristics, degree of polymerization, and volume fractions of the topological polymers used in this study.

Polymer	*M_n,NMR_^a^* (g/mol)	*M_w_/M_n_^b^*	Non-Polar Block				Polar Block			
			*DP_NB_^c^*	*ϕ_NB_^d^*	*ρ_e_^e^* (nm^−3^)	*ρ_m_^f^* (g/cm^3^)	*DP_PB_^c^*	*ϕ_PB_^d^*	*ρ_e_^e^* (nm^−3^)	*ρ_m_^f^* (g/cm^3^)
*c*-PDGE-*b*-PTEGGE	22,300	1.04	50	0.52	310	0.91	51	0.48	-	0.96 *^g^*
*l*-PDGE-*b*-PTEGGE	21,900	1.04	49	0.49	341	1.01	51	0.51	353	1.05
*l*-PPCGE-*b*-PDDGE	12,100	1.10	30	0.70	380 *^h^*	1.14	10	0.30	405 *^i^*	1.24

*^a^* Number-average molecular weight of polymer determined by ^1^H nuclear magnetic resonance (NMR) spectroscopy analysis. *^b^* Polydispersity index of polymer determined by gel permeation chromatography analysis in tetrahydrofuran (THF, polystyrene standard was used). *^c^* Number-average degree of polymerization of non-polar or polar block determined by ^1^H NMR spectroscopy analysis. *^d^* Volume fraction of non-polar or polar block estimated from the *M_n,NMR_* and *ρ_m_* data. *^e^* Electron density of non-polar or polar homopolymer in films determined by X-ray reflectivity analysis. *^f^* Mass density of non-polar or polar homopolymer in films obtained from the electron density determined by X-ray reflectivity analysis. *^g^* Cyclic PTEGGE homopolymer is assumed to have *ρ_m_* = 0.96 g/cm^3^ [32]. *^h^* The *ρ_m_* of PDDGE homopolymer is assumed to be same with that of poly(oxy(*n*-dodecylthiomethyl)ethylene) [36]. *^i^* The *ρ_m_* of PPCGE homopolymer is assumed to be same with that of poly(oxy(11-phosphorylcholineundecylthiomethyl)ethylene) [36].

**Table 2 polymers-10-01347-t002:** Structural parameters of the normal *c*-PDGE-*b*-PTEGGE micelle formed in a mixture of 75 wt % ethanol and 25 wt % water, which were analyzed by DLS and X-ray scattering analyses.

DLS Analysis	Peaks Appeared in Size Distribution										
	Peak 1										
	*R*_*h*,1_*^a^* (nm)	*φ_1_^b^* (%)					*R_h,z_^c^* (nm)	*PDI_DLS_^d^*	*R_g,TPS_/R_h,z_*	*R_g,TPS_/R* _*h*,1_	
Intensity-weighted distribution	21 (5) *^e^*	100					20	0.03	0.42	0.40	
Number-weighted distribution	17 (4)	100								0.51	
Volume-weighted distribution	19 (4)	100								0.46	
X-ray scattering analysis	*R_g,G_^f^* (nm)	*R_g,IFT_^g^* (nm)	*R_g,TPS_^h^* (nm)	*R_max_^i^* (nm)	*R_p_^j^* (nm)	*R_e_^k^* (nm)	*D_max_*^l^ (nm)	*R_g,G_/R_g,IFT_*	*R_max_/R_g,IFT_*	*D_max_/R_max_*	ε*_el_^m^*
Guinier	7.75										
Indirect Fourier transformation (IFT)		7.97		10.58			23.00	0.97	1.33	2.17	
Three-phase ellipsoid			8.50		9.70	11.55					0.84

*^a^* Averaged hydrodynamic radius. *^b^* Fraction in percent. *^c^ z*-Averaged hydrodynamic radius. *^d^* Polydispersity index in the DLS data analysis, which is defined by Equation (12). *^e^* Standard deviation. *^f^* Radius of gyration determined from Guinier analysis. *^g^* Radius of gyration determined from IFT analysis. *^h^* Radius of gyration determined from three phase ellipsoid (which is a three phase structure (TPS)) analysis. *^i^* Radius determined from the peak maximum of the *p*(*r*) function in IFT analysis. *^j^* Polar radius. *^k^* Equatorial radius. *^l^* Maximum dimension determined from the *p*(*r*) function in IFT analysis. *^m^* Ellipsoidicity (= polar radius/equatorial radius).

**Table 3 polymers-10-01347-t003:** Structural parameters of the normal *l*-PDGE-*b*-PTEGGE micelle formed in a mixture of 75 wt % ethanol and 25 wt % water, which were analyzed by DLS and X-ray scattering analyses.

DLS Analysis	Peaks Appeared in Size Distribution										
	Peak 1		Peak 2		Peak 3						
	*R*_*h*,1_*^a^* (nm)	*ϕ*_1_*^b^* (%)	*R*_*h*,2_ (nm)	*ϕ*_2_ (%)	*R*_*h*,3_ (nm)	ϕ*_3_* (%)	*R_h,z_^c^* (nm)	*PDI_DLS_^d^*	*R_g,TPS_/R_h,z_*	*R_g,TPS_/R* _*h*,1_	
Intensity-weighted distribution	78 (17) *^e^*	15.0	467 (99)	81.9	2780 (0)	3.1	1003	1.00	0.02	0.29	
Number-weighted distribution	62 (15)	96.6	430 (101)	3.4		0				0.36	
Volume-weighted distribution	71 (18)	5.2	488 (112)	59.9	2795 (290)	34.9				0.31	
X-ray scattering analysis	*R_g,G_^f^* (nm)	*R_g,IFT_^g^* (nm)	*R_g,TPS_^h^* (nm)	*R_max_^i^* (nm)	*R_p_^j^* (nm)	*R_e_^k^* (nm)	*D_max_*^l^ (nm)	*R_g,G_/R_g,IFT_*	*R_max_/R_g,IFT_*	*D_max_/R_max_*	ε*_el_^m^*
Guinier	18.57										
IFT		18.05		22.42			59.00	1.03	1.24	2.63	
Three-phase ellipsoid			22.20		34.50	23.00					1.50

*^a^* Averaged hydrodynamic radius. *^b^* Fraction in percent. *^c^ z*-Averaged hydrodynamic radius. *^d^* Polydispersity index in the DLS data analysis, which is defined by Equation (12). *^e^* Standard deviation. *^f^* Radius of gyration determined from Guinier analysis. *^g^* Radius of gyration determined from IFT analysis. *^h^* Radius of gyration determined from three phase ellipsoid (a three phase structure (TPS)) analysis. *^i^* Radius determined from the peak maximum of the *p*(*r*) function in IFT analysis. *^j^* Polar radius. *^k^* Equatorial radius. *^l^* Maximum dimension determined from the *p*(*r*) function in IFT analysis. *^m^* Ellipsoidicity (= polar radius/equatorial radius).

**Table 4 polymers-10-01347-t004:** Structural parameters of the reverse *l*-PPCGE-*b*-PDDGE micelle formed in chloroform, which were analyzed by DLS and X-ray scattering analyses.

DLS Analysis	Peaks Appeared in Size Distribution										
	Peak 1		Peak 2		Peak 3						
	*R*_*h*,1_*^a^* (nm)	*ϕ*_1_*^b^* (%)	*R*_*h*,2_ (nm)	*ϕ*_2_ (%)	*R*_*h*,3_ (nm)	*ϕ*_3_ (%)	*R_h,z_^c^* (nm)	*PDI_DLS_^d^*	*R_g,TPS_/R_h,z_*	*R_g,TPS_/R* _*h*,1_	
Intensity-weighted distribution	8 (2) *^e^*	3.5	42 (18)	96.5			32	0.22	0.26	1.07	
Number-weighted distribution	7 (2)	100								1.30	
Volume-weighted distribution	7 (2)	69.6	25 (12)	30.4						1.17	
X-ray scattering analysis	*R_g,G_^f^* (nm)	*R_g,IFT_^g^* (nm)	*R_g,TPS_^h^* (nm)	*R_max_^i^* (nm)	*R_p_^j^* (nm)	*R_e_^k^* (nm)	*D_max_^l^* (nm)	*R_g,G_/R_g,IFT_*	*R_max_/R_g,IFT_*	*D_max_/R_max_*	ε*_cyl_^m^*
Guinier	8.44										
IFT		8.61		9.95			27.30	0.98	1.16	2.74	
Three-phase cylinder			8.43		7.70	22.30					1.45

*^a^* Averaged hydrodynamic radius. *^b^* Fraction in percent. *^c^ z*-Averaged hydrodynamic radius. *^d^* Polydispersity index in the DLS data analysis, which is defined by Equation (11). *^e^* Standard deviation. *^f^* Radius of gyration determined from Guinier analysis. *^g^* Radius of gyration determined from IFT analysis. *^h^* Radius of gyration determined from three phase cylinder (a three phase structure (TPS)) analysis. *^i^* Radius determined from the peak maximum of the *p*(*r*) function in IFT analysis. *^j^* Radius of cylindrical micelle. *^k^* Height of cylindrical micelle. *^l^* Maximum dimension determined from the *p*(*r*) function in IFT analysis. *^m^* Aspect ratio (ε*_cyl_* = cylinder height/diameter).

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
