# Peer review of "A Comparative Study of Dynamic Light and X-ray Scatterings on Micelles of Topological Polymer Amphiphiles"

_polymers, 2018, doi:10.3390/polym10121347_

Round 1

Reviewer 1 Report

The manuscript "a comperative study of dynamic light and x-ray scattering on micelles of topological polymer amphiphiles" is a thorough study on several polymeric micelles characterized by DLS and SAXS that surely deserves publication. Especially the ring polymers are rarely used before and the overall choice of monomers is quite rare. So I support the publication.

My main concern is the heavy use of DLS and SAXS in the literature makes the detailed repetition of the lengthy presentation of the background appear rather educational than scientifically necessary to the reader. I would also like to ask the editor what the habits of the journal are. Of course all formulae are correct and do not contradict the scientific results.

The second point is the disagreement between DLS and SAXS. This is also known for a long time. Maybe the expressions could appear less surprised than in the current version. 

I cannot make real comments to the language, since I am not native. If there is a second referee, I would hope that he/she can make his/her comments.

Author Response

We thank this reviewer very much for the comments and suggestions on the manuscript. Further
we thank the reviewer for the recommendation to publish the manuscript in the POLYMERS:
Reviewer’s statement – “The manuscript "a comparative study of dynamic light and X-ray scattering on micelles of topological polymer amphiphiles" is a thorough study on several polymeric micelles
characterized by DLS and SAXS that surely deserves publication. Especially the ring polymers are rarely used before and the overall choice of monomers is quite rare. So I support the publication.

Reviewer 2 Report

In this comparative study, authors have used dynamic light and x-ray scattering to evaluate micelles of three different diblock copolymers with either linear or cyclic topologies. Considering the magnitude of literature, using DLS data, like a black box, throwing some numbers in nanometres, this is an important work that will force the community to pause and think about the numbers one gets out of these systems! Having said that there is no justification for why certain things were done in certain fashion. I strongly feel that the authors should carefully take stock of key strengths in their data set and restructure the manuscript. Certainly they will need to further strengthen their studies with electron micrscopy.

To me, only having polymer a and b (figure 1) makes sense for comparison. As for polymer c, either the justifications are not included in the current version or its just random polymer added into this manuscript… in either case, as a reviewer, it’s not convincing.

Following pointers might help authors in revising the manuscript.

1.       Introduction needs major revision. In the current format it is vacuous too broad with too many references!

2.       Neither in the introduction nor in the manuscript there are no major justifications as to why the comparative studies were restricted to DLS and X-ray scattering (and why not SANS?)

3.       For the work to be considered for publication at least authors must have transmission electron micrographs (TEM). Even better would be have cryogenic-TEM

4.       What is the rational for the polymers used? Polymer a and b (Figure 1) are excellent comparison. As for polymer c, no justification as to why this was used. Considering polymers a and b are near perfect comparison, why was polymer c with different composition, amphiphilic balance was used? It is well known that the micellar morphology is very sensitive to the amphiphilic balance and composition. Lack of clear and strong justification for these specific three polymers,  undermines the confidence in the work

5.       Why was data set in different solvents were compared ( 75% ethanol and 25 p% water versus chloroform section 2.2)? As solvent has a bearing on the self-assembly outcome, to me these are not comparable.

6.       Section 2.4 and similar sections, entirely new? If this is from manual or prior work etc., please sort it out. If these are new, you may want to consider, including them as an appendix.

7.       Manuscript must be thoroughly checked for typos etc. For instance, page 24, last line – guess should be ratios.

8.       Authors must consider trimming down the references (76 for a short communication, seems bit odd). Citations like 1 – 12; 18-33 and 3448, for a general statements seems excessive.

Author Response

We thank this reviewer very much for the comments and suggestions on the manuscript:
Reviewer’s statement – “In this comparative study, authors have used dynamic light and x-ray
scattering to evaluate micelles of three different diblock copolymers with either linear or cyclic topologies. Considering the magnitude of literature, using DLS data, like a black box, throwing some numbers in nanometres, this is an important work that will force the community to pause and think about the numbers one gets out of these systems! Having said that there is no justification for why certain things were done in certain fashion. I strongly feel that the authors should carefully take stock of key strengths in their data set and restructure the manuscript. Certainly they will need to further strengthen their studies with electron microscopy.”

Reviewer 3 Report

The paper “A comparative study of dynamic light and X-ray scatterings on micelles of topological polymer amphiphiles” by Ree et al. investigates the differences in the micellar characteristics, such as size and size distributions, obtained by x-ray scattering and DLS.

In general the paper is well written, the conclusions are supported by the results but there are some important drawbacks. The most important is that some results obtained by x-ray for the cyclic and linear PDGE-b-PTEGGE were already published in “Well-defined and stable nanomicelles self-assembled from brush cyclic and tadpole copolymer amphiphiles:a versatile smart carrier platform”, NPG Asia Materials, 9, e453; doi:10.1038/am.2017.205. Even if this article was cited by the authors, I think that these published results must be discussed in more detail in the introduction section. Moreover, the originality of the present study must be highlighted with respect to the already published results.

Also concerning the introduction section, I think that this section must be modified. First of all, the authors must provide some general references (revue articles) concerning the micellization or self-assembly of amphiphilic copolymers in aqueous and organic media. Two recent references are:

-          Micellization of synthetic and polysaccharides-based graft copolymers in aqueous media”, Prog. Polym. Sci., 73, 32-60, 2017

-          Self-Assembly of block and graft copolymers in organic solvents: An overview of recent advances”, Polymers, 10, 62, 2018

Then, they must also provide and discussed studies concerning the micellization of other cyclic amphiphilic copolymers.  

It should be of interest for the readers to know why the authors have chosen a 75/25 wt/wt% ethanol/water ratio for the preparation of the micellar solutions?! At this point, I need to remark that this method of preparing micellar solutions is not recommended as large and instable aggregates are generally obtained. In the “micelle formation” section the authors must provide more information concerning the time of solution preparation and also the time after which the micellar solutions were analyzed.  As the large aggregates obtained by the precipitation method are not stable, I suppose that an evolution in time of the micellar characteristics occurs. Moreover, the authors must provide the value of the viscosity for the mixture ethanol/water which was used for the DLS measurements.

Concerning the DLS measurements, I would like to known if the authors have tried to analyze unfiltered micellar solutions?!  It is quite curious that they observed in DLS sizes higher than 400 nm when all the solutions were filtered with filters having a pore size of 200 nm…Moreover, the solutions were analyzed immediately after the filtration or an equilibration time under agitation was used?! Furthermore, the authors provide a standard deviation but they not indicate the number of measurements carried out…Standard deviation values are given in nm?! A standard deviation must also be provided for the Rh,z.

The authors provide for the cyclic copolymer  a PDIDLS  value of 0.03 but from the figure 2 it seems that the polydispersity might be much higher… For the sake of clarity in fig 2, 4 and 6 should be provided the same type of representation, either having a logarithmic scale or not but not the both.

For the linear PDGE-b-PTEGGE a PDI value equal to 1 was obtained. However, it is well known that in this case, where the system is very polydisperse, the values for Rh,volume and Rh,number are not significant and should not be taken into account. Generally, this is valid for all the systems where PDI>0.5.

Finally, the authors state that the DLS analysis is applicable only for spherical particles but they did not study such type of micelles…therefore, it would be of interest if the authors could analyze spherical micelles before drawing such conclusions.

In view of the above, I recommend the publication of this manuscript in Polymers only after major corrections.

Author Response

We thank this reviewer very much for the comments and suggestions on the manuscript. Our
manuscript has been revised according to the comments of the reviewer. The following section
contains our replies to the comments of the reviewer.

Round 2

Reviewer 2 Report

I find that the authors have satisfactorily addressed the comments. Particularly including rationale for having these three specific polymers, improved the clarity. I would like to thank the authors for addressing the concerns.

Only minor concern is - I am not sure what they mean by - molecular micelles and molecular vesicles in the introduction. Is this is to distinguish from unimolecular namostructures (say dendrimeric amphiphile) versus self-assembled version? In any case it would be better to fix this before publication.

Author Response

We thank the reviewer very much for the comment.
We are sorry about such confusion. Our intension was to distinguish the micelles and
vesicles made of small organic molecules from the micelles and vesicles made of polymers.
To avoid any further confusions, they were replaced by “(i) small-molecule-based micelles
and vesicles,….” We hope that the reviewer accepts the revised those; otherwise please give
us a better word set.

Reviewer 3 Report

can be published in the present form.

Author Response

We thank this reviewer very much for reviewing the revised manuscript.